# Rate of Intensive Care Unit admission and outcomes among patients with coronavirus: A systematic review and Meta-analysis

**Semagn Mekonnen Abate[1]\*, Siraj Ahmed Ali[1], Bahiru Mantfardo[2], Bivash Basu[3]**

**1** Department of Anesthesiology, College of Health Sciences and Medicine, Dilla University, Dilla, Ethiopia,
**2** Department of Internal Medicine, College of Health Sciences and Medicine, Dilla University, Dilla, Ethiopia,
**3** Department of Anesthesiology, College of Medicine, University of Calcutta, Calcutta, India

\* semmek17@gmail.com

**Data Availability Statement:** The metadata can be accessed by https://doi.org/10.6084/m9.figshare.12489701.v1.

## Abstract

### Background

The rate of ICU admission among patients with coronavirus varied from 3% to 100% and the mortality was as high as 86% of admitted patients. The objective of the systematic review was to investigate the rate of ICU admission, mortality, morbidity, and complications among patients with coronavirus.

### Methods

A comprehensive strategy was conducted in PubMed/Medline; Science direct and LILACS from December 2002 to May 2020 without language restriction. The Heterogeneity among the included studies was checked with forest plot, $\chi^2$ test, I2 test, and the p-values. All observational studies reporting rate of ICU admission, the prevalence of mortality and its determinants among ICU admitted patients with coronavirus were included and the rest were excluded

### Result

A total of 646 articles were identified from different databases and 50 articles were selected for evaluation. Thirty-seven Articles with 24983 participants were included. The rate of ICU admission was 32% (95% CI: 26 to 38, 37 studies and 32, 741 participants). The Meta-Analysis revealed that the pooled prevalence of mortality in patients with coronavirus disease in ICU was 39% (95% CI: 34 to 43, 37 studies and 24, 983 participants).

### Conclusion

The Meta-Analysis revealed that approximately one-third of patients admitted to ICU with severe Coronavirus disease and more than thirty percent of patients admitted to ICU with a severe form of COVID-19 for better care died which warns the health care stakeholders to give attention to intensive care patients.

**Funding:** The authors received no specific funding for this work.

**Competing interests:** The authors declare that there are no competing interests.

## Registration

This Systematic review and Meta-Analysis was registered in Prospero international prospective register of systemic reviews (CRD42020177095) on April 9/2020.

## 1. Introduction

The Coronavirus belongs to large groups of viruses that cause serious health problems affecting the respiratory, gastrointestinal, liver, and central nervous system of humans, livestock, Bats, mice, and other wild animals [1–6]. The infection mainly affects the respiratory system and manifested with fever, dry cough, and difficulty breathing. In the late stages of the infection, the patient may die due to pneumonia and acute respiratory distress syndrome [4, 7–10].

The Severe acute respiratory syndrome (SARS-CoV-2) novel coronavirus was identified in Wuhan, Hubei province of China in December 2019 by the Chinese Center for Disease and Prevention from the throat swab of a patient and the virus is named severe acute respiratory distress COV-2 by WHO which causes Coronaviruses disease 2019 (COVID-19) [11, 12].

The clinical manifestation of the current coronavirus infection is similar to Severe acute respiratory syndrome (SARS-CoV) outbreak that occurred in the Guangdong Province of China by the year 2002–2003 [13–16] and another novel human coronavirus called Middle East Respiratory Syndrome-CoV (MERS-CoV) which was identified in the Middle East and other Arabian regions in 2012 [17–20].

The World Health Organization (WHO) is named the current virus as severe acute respiratory distress COV-2 which causes coronaviruses disease 2019 (COVID-19). The WHO has declared the novel coronavirus (COVID-19) outbreak as a global pandemic on March 11, 2020 [21].

Globally, More than 5 million confirmed cases and 400, 000 deaths were reported by the World Health Organization (WHO) as of June 9, 2020 [22]. The American region accounted for the highest number of cases and deaths which was more than 3 million and 200,000 respectively. The European region accounted for the second-highest confirmed cases and death which were more than 2 million confirmed cases and 183 thousand deaths. The total number of confirmed cases and death in the Eastern Mediterranean region accounted for approximately 660, 000, and 15,000 respectively [22].

The number of laboratory-confirmed cases and deaths in the African region was the lowest for the last couple of months but the rate of spreading in this region is increasing at an alarming rate and expected to be very high in the next couple of months if it continues as this rate. The current report in Ethiopia is very small which is 2500 confirmed cases and 27 deaths but there are many cases in short periods which is more than150 cases per day [22]. It is estimated that the number even may be very high because the diagnosis is limited only in the capital.

The challenge of COVID-19 is very high globally due to a lack of proven treatment and the complexity of its transmission [12, 19, 23–28]. However, it will be more catastrophic for low and middle-income countries because of very poor health care system, high illiteracy and low awareness of the disease and its prevention, lack of skilled health personnel, scarce Intensive Care Unit, a limited number of mechanical ventilators and prevalence of co-morbidities/infection along with malnutrition.

The severity of the disease is depending on several factors. Studies showed that patients with co-morbidities including (Asthma, COPD, Tuberculosis, Pneumonia, Acute respiratory distress syndrome (ARDS), Diabetes mellitus, hypertension, renal disease, hepatic disease, and

cardiac disease), history of smoking, and history of substance use, male gender and age greater than 60 years were more likely to die or develop undesirable outcomes [25, 28–35].

The outcomes of patients with coronavirus infection are very variable. Studies also showed that the rate of ICU admission among coronavirus infected patients was higher which ranged from 3% to 100% of confirmed cases [14, 17, 19, 26, 28, 36–39]. Studies also showed that the prevalence of mortality among intensive care patients with coronavirus infection was very high which ranged from 6% to 86% of admitted patients [14, 17, 19, 26, 28, 36–39].

The global rate of ICU admission, the prevalence of mortality, comorbidities, complication, number of cases demanding mechanical ventilator, length of stay and independent risk factors for ICU mortality are very important variables to be determined to reduce patient mortality and morbidity through varies mitigating strategies including but not limited to increasing number of ICU beds, mechanical ventilator, skilled professionals, integrated monitors and reducing possible risk factors. Therefore, the objectives of this systematic review and Meta-Analysis was to provide global evidence on the rates of ICU admission, the prevalence of mortality, comorbidity, complications, and independent risk factors of mortality among patients with COVID-19 admitted in ICU.

## 2. Materials and methods

### 2.1. Protocol and registration

The systematic review and meta-analysis were conducted based on the Preferred Reporting Items for Systematic and Meta-analysis (PRISMA) protocols [40]. This Systematic Review and Meta-Analysis was registered in Prospero international prospective register of systemic reviews (CRD42020177095) on April 9/2020.

### 2.2. Inclusion and exclusion criteria

**2.2.1. Inclusion criteria.** All observational (case series, cross-sectional, cohort, and case-control) studies reporting rate of ICU admission, the prevalence of mortality, morbidity, complication, and its determinants among ICU admitted patients with coronavirus (SARS-COV, MERS and SARS-COV 2) were included.

**2.2.2. Exclusion criteria.** Studies that didn't report the rate of ICU admission, the prevalence of ICU mortality, and risk factors among patients with coronavirus were excluded. Besides, Randomized controlled trials, case-control studies, Systemic reviews, and Case reports were excluded.

### 2.3. Outcomes of interest

**2.3.1. Primary outcomes.** The primary outcome of interest was rates of ICU admission and mortality among patients admitted with Coronaviruses during SARS, MERS, and COVID-19 pandemic.

**2.3.2. Secondary outcomes.** Prevalence of morbidity, the prevalence of complication, and its determinants among patients admitted with Coronaviruses during SARS, MERS, and COVID-19 pandemic.

### 2.4. Search strategy

The search strategy was intended to explore all available published and unpublished studies among Coronaviruses infected patients admitted to ICU from December 2002 to May 2020 without language restrictions. A comprehensive initial search was employed in PubMed/Medline, Science direct, and LILACS followed by an analysis of the text words contained in Title/

Abstract and indexed terms. A second search was undertaken by combining free text words and indexed terms with Boolean operators. The third search was conducted with the reference lists of all identified reports and articles for additional studies. Finally, an additional and grey literature search was conducted on Google scholars. The PubMed/Medline database was searched with the following terms: SARS[Title/Abstract]) OR (SARS-COV-2[Title/Abstract])) OR (COVID-19[Title/Abstract])) AND (MERS[Title/Abstract])) AND (mortality[Title/Abstract])) OR (morbidity[Title/Abstract])) AND (ICU[Title/Abstract])) OR (hospital[Title/Abstract])) AND (prevalence[Title/Abstract])) AND (risk factors[Title/Abstract])).

### 2.5. Data extraction

The data from each study were extracted with two independent authors with a customized format. The disagreements between the two independent authors were resolved by the other two authors. The extracted data included: Author names, country, date of publication, sample size, the rates of ICU admission, mortality, types of Coronavirus, types of comorbidity, complications, and risk factors. Finally, the data were then imported for analysis in R software version 3.6.1 and STATA 14.

### 2.6. Assessment of methodological quality

Articles identified for retrieval were assessed by two independent Authors for methodological quality before inclusion in the review using a standardized critical appraisal Tool adapted from the Joanna Briggs Institute [45,46] (S1 Table). The disagreements between the Authors appraising the articles were resolved through discussion with the other Two Authors. Articles with average scores greater than fifty percent were included for data extraction.

### 2.7. Data analysis

Data analysis was carried out in R statistical software version 3.6.1 and STATA 14. The pooled rates of ICU admission and prevalence of mortality, comorbidity, complication among corona virus-infected patients were determined with a random effect model as there was substantial heterogeneity between the included studies. The Heterogeneity among the included studies was checked with forest plot, $\chi2$ test, $I^2$ test, and the p-values. Subgroup analysis was conducted by Country, type of coronavirus, types of comorbidity, and complications. Publication bias was checked with a funnel plot and the objective diagnostic test was conducted with Egger's correlation, Begg's regression tests, and Trim and fill method. Furthermore, moderator analysis was carried out to identify the independent predictors of ICU mortality among corona cases. The results were presented based on the Preferred Reporting Items for Systemic Reviews and Meta-Analysis (PRISMA) [40].

### 2.8. Ethics approval and consent to participate

Ethical clearance and approval were obtained from the ethical review board of the College of Health Science and Medicine.

## 3. Results

### 3.1. Selection of studies

A total of 646 articles were identified from different databases with an initial search. Fifty articles were selected for evaluation after the successive screening. Thirty-seven Articles with 24983 participants were included in the systematic review and Meta-Analysis while thirteen studies were excluded with reasons (Fig 1).

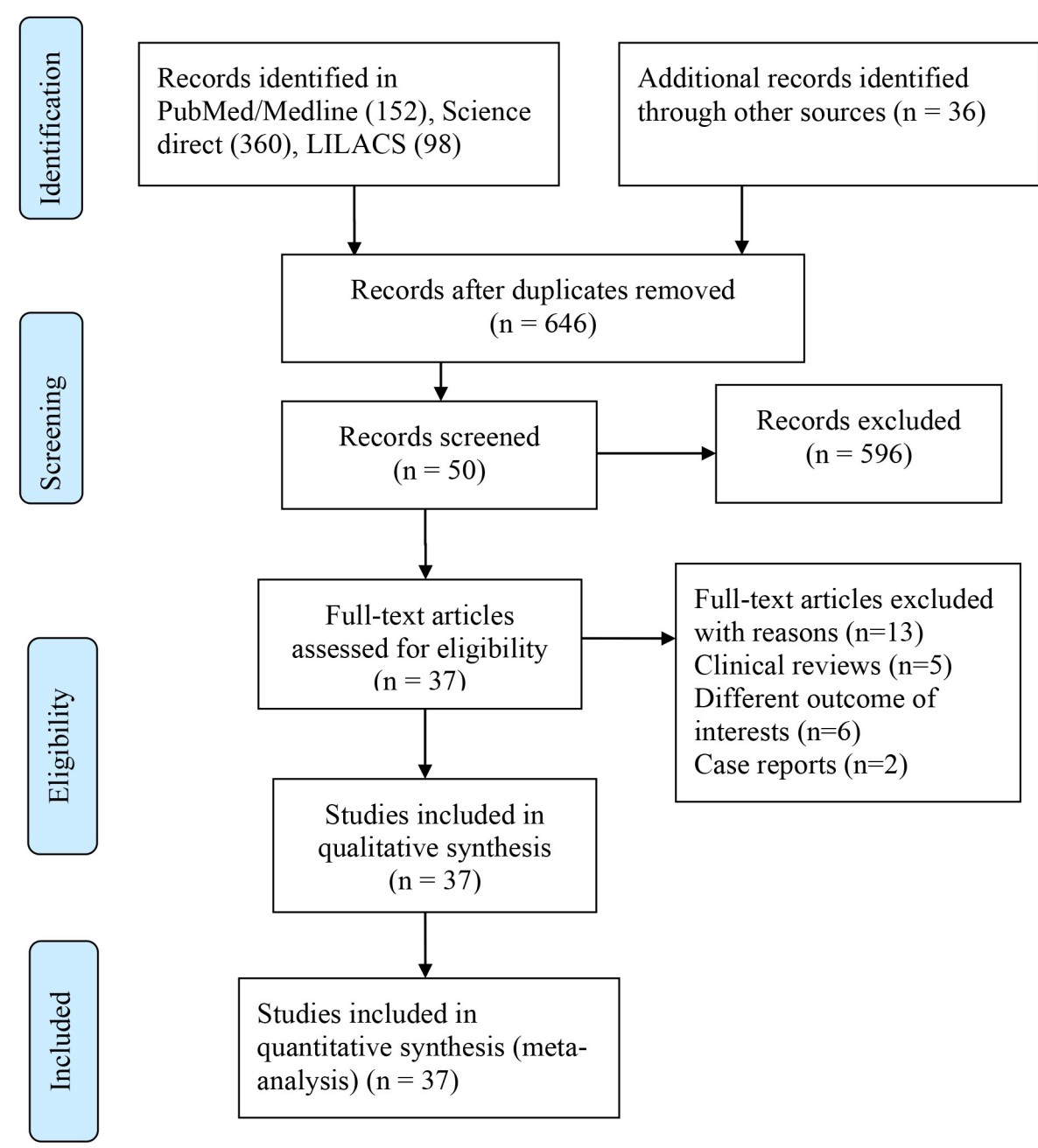

**Fig 1. Prisma flow chart.**

## 3.2. Characteristics of included studies

Thirty-seven studies conducted on Coronavirus reporting rates of ICU admission and patient outcomes with 24983 participants were included (Table 1). Thirteen studies were excluded with reasons (S1 Table). The methodological quality of included studies was moderate to high quality as depicted with the Joanna Briggs Appraisal tool for observational studies (S2 Table).

Twenty-six of the included studies were conducted on a newly emerged Coronavirus (SARS-CoV-2), COVID-19. Seven studies were conducted during and after the aftermath of

**Table 1. Methodological quality of included studies.**

| Author(s) | Year | Event | Sample | Country | Types of Coronavirus | Quality Score | Prevalence (95% CI) |
|-----------|------|-------|--------|---------|----------------------|---------------|---------------------|
| Liu et al[41] | 2020 | 7 | 11 | China | SARS-COV-2 | 8 | 64(31, 89) |
| Xu et al[42] | 2020 | 1 | 2 | China | SARS-COV-2 | 6 | 50(1, 99) |
| Arentz et al[37] | 2020 | 11 | 17 | USA | SARS-COV-2 | 5 | 65(38, 86) |
| Bhatraju et al[43] | 2020 | 12 | 24 | USA | SARS-COV-2 | 5 | 50(29, 71) |
| Bialek et al[44] | 2020 | 55 | 121 | USA | SARS-COV-2 | 5 | 45(36, 55) |
| Cao et al[45] | 2020 | 3 | 4 | China | SARS-COV-2 | 4 | 75(19, 99) |
| Chen et al[46] | 2020 | 2 | 22 | China | SARS-COV-2 | 6 | 9(1,29) |
| Chen et al[14] | 2020 | 11 | 23 | China | SARS-COV-2 | 8 | 48(27, 69) |
| Huang et al[47] | 2020 | 6 | 13 | China | SARS-COV-2 | 6 | 46(19, 75) |
| Petrilli et al[48] | 2020 | 116 | 457 | USA | SARS-COV-2 | 6 | 25(21, 30) |
| Richardson et al[49] | 2020 | 18 | 373 | USA | SARS-COV-2 | 7 | 5(3, 8) |
| Simonnet et al[50] | 2020 | 18 | 124 | France | SARS-COV-2 | 5 | 15(9, 22) |
| Wang et al[51] | 2020 | 6 | 36 | China | SARS-COV-2 | 6 | 17(6, 33) |
| Wu et al[52] | 2020 | 44 | 53 | China | SARS-COV-2 | 6 | 83(70,72) |
| Yang et al[28] | 2020 | 32 | 52 | China | SARS-COV-2 | 6 | 62(47, 75) |
| Young et al[6] | 2020 | 1 | 2 | Singapore | SARS-COV-2 | 6 | 50(1, 99) |
| Guan et al[53] | 2020 | 15 | 1099 | China | SARS-COV-2 | 6 | 1(1, 2) |
| Zhou et al[54] | 2020 | 39 | 50 | China | SARS-COV-2 | 6 | 78(64, 88) |
| Lodigiania et al[55] | 2020 | 8 | 62 | Italy | SARS-COV-2 | 7 | 13(6, 24) |
| Kloka et al[56] | 2020 | 41 | 184 | Holland | SARS-COV-2 | 5 | 22(16, 29) |
| Lei et al [57] | 2020 | 7 | 15 | China | SARS-COV-2 | 6 | 47(21, 73) |
| Docherty et al[58] | 2020 | 3001 | 20133 | UK | SARS-COV-2 | 6 | 15(14, 15) |
| Du et al [59] | 2020 | 6 | 51 | China | SARS-COV-2 | 5 | 12(4, 24) |
| Ling et al[60] | 2020 | 8 | 49 | China | SARS-COV-2 | 5 | 16(7, 30) |
| Zangrillo et al [61] | 2020 | 14 | 61 | Italy | SARS-COV-2 | 4 | 23(13, 35) |
| Grasselli et al [62] | 2020 | 405 | 1591 | Italy | SARS-COV-2 | 6 | 25(23, 28) |
| Chan et al[13] | 2003 | 18 | 39 | China | SARS-COV | 7 | 46(30, 63) |
| Chen et al[12] | 2005 | 21 | 33 | Taiwan | SARS-COV | 5 | 64(45, 80) |
| Choi et al[15] | 2003 | 32 | 69 | China | SARS-COV | 8 | 46(34, 59) |
| Lew TW et al[63] | 2003 | 20 | 46 | Singapore | SARS-COV | 8 | 43(29, 59) |
| Almekhlafie et al[64] | 2016 | 23 | 27 | Saudi Arabia | MERS-CoV | 6 | 85(66, 96) |
| Al-Hameed et al[18] | 2016 | 5 | 8 | Saudi Arabia | MERS-CoV | 6 | 63(24, 91) |
| Garbati et al[65] | 2016 | 1 | 4 | Saudi Arabia | MERS-CoV | 8 | 25(1, 81) |
| Al Ghamdi et al[66] | 2016 | 19 | 37 | Saudi Arabia | MERS-CoV | 5 | 51(34, 68) |
| Halim et al[26] | 2016 | 14 | 32 | Saudi Arabia | MERS-CoV | 7 | 44(26, 62) |
| Saad et al[33] | 2014 | 42 | 49 | Saudi Arabia | MERS-CoV | 8 | 86(73, 94) |
| Arabi YM et al[19] | 2014 | 5 | 10 | Saudi Arabia | MERS-CoV | 6 | 50(19, 81) |

Q: question; Y: yes; N: No

the Middle East respiratory syndrome epidemic in the Middle East and other Arabian regions in 2012 while the remaining four studies were conducted during the severe acute respiratory syndrome (SARS-CoV) outbreak in China in 2002.

The included studies were conducted in different regions of the world. Sixteen studies were conducted in China, seven studies in Saudi Arabia, five studies in the United States of America, three studies in Italy, two studies in Singapore, one study in Holland, the United Kingdom, and France.

All of the included studies reported rates of ICU admission and outcomes of patients while staying in ICU. The majority of the included studies reported the presence of comorbidities and complications in ICU such as death, acute respiratory distress syndrome, renal failure, shock, and discharge.

### 3.3. Meta-analysis

**3.3.1. Rate of ICU admission.** Thirty-seven studies reported ICU admission were included for Meta-analysis. The number of ICU admission was taken for estimation of pooled prevalence of mortality instead of the total sample size because we wanted to know the number of ICU deaths from those Admitted in ICU. However, the rates of ICU admission were estimated with the total sample size. The pooled rate of ICU admission was 32% (95% CI: 26 to 38, 37 studies and 32, 741 participants) (Fig 2).

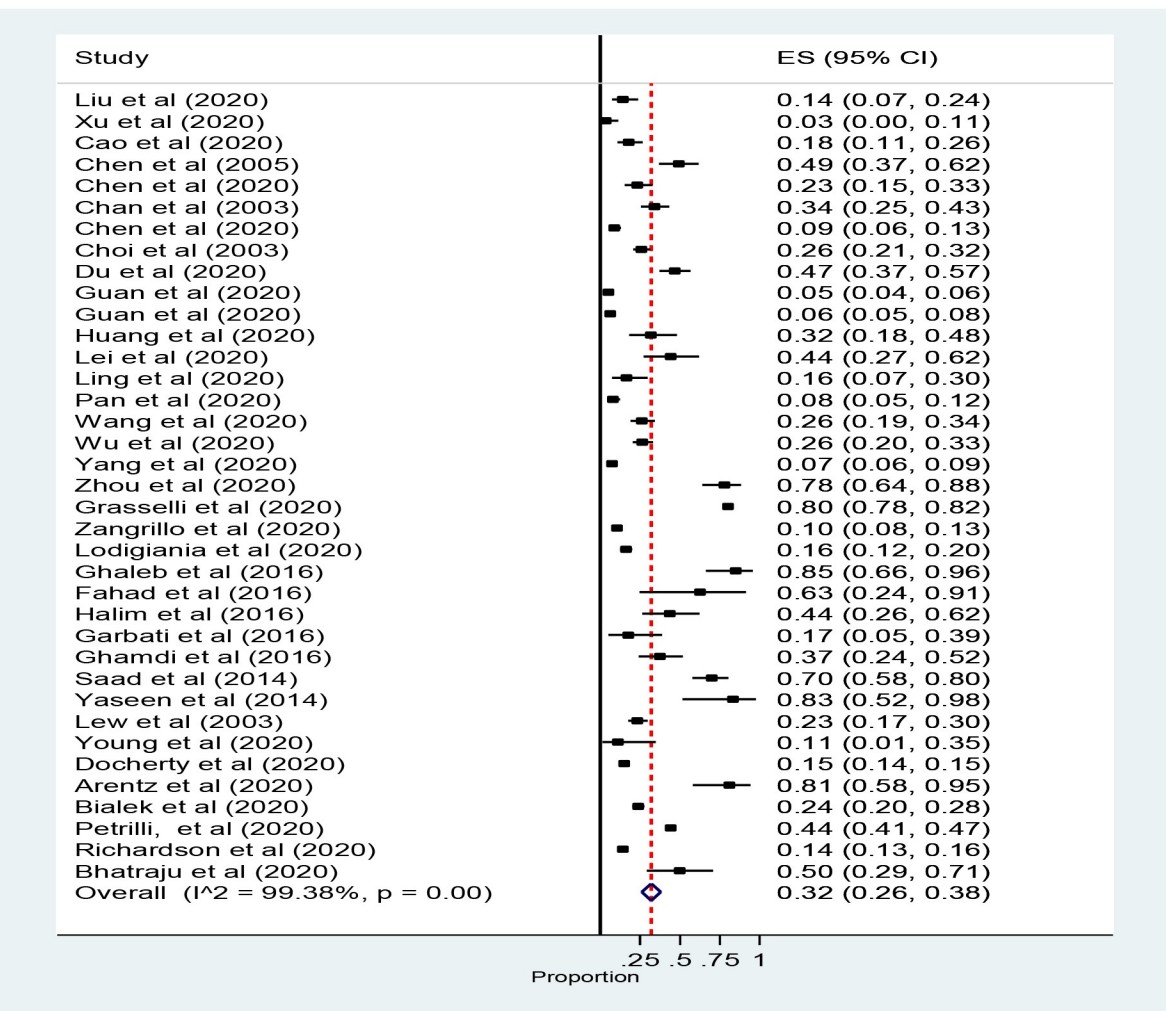

**Fig 2. Forest plot for the prevalence of ICU admission patients with coronavirus: The midpoint of each line illustrates the prevalence; the horizontal line indicates the confidence interval, and the diamond shows the pooled prevalence.** ICU: Intensive Care Unit.

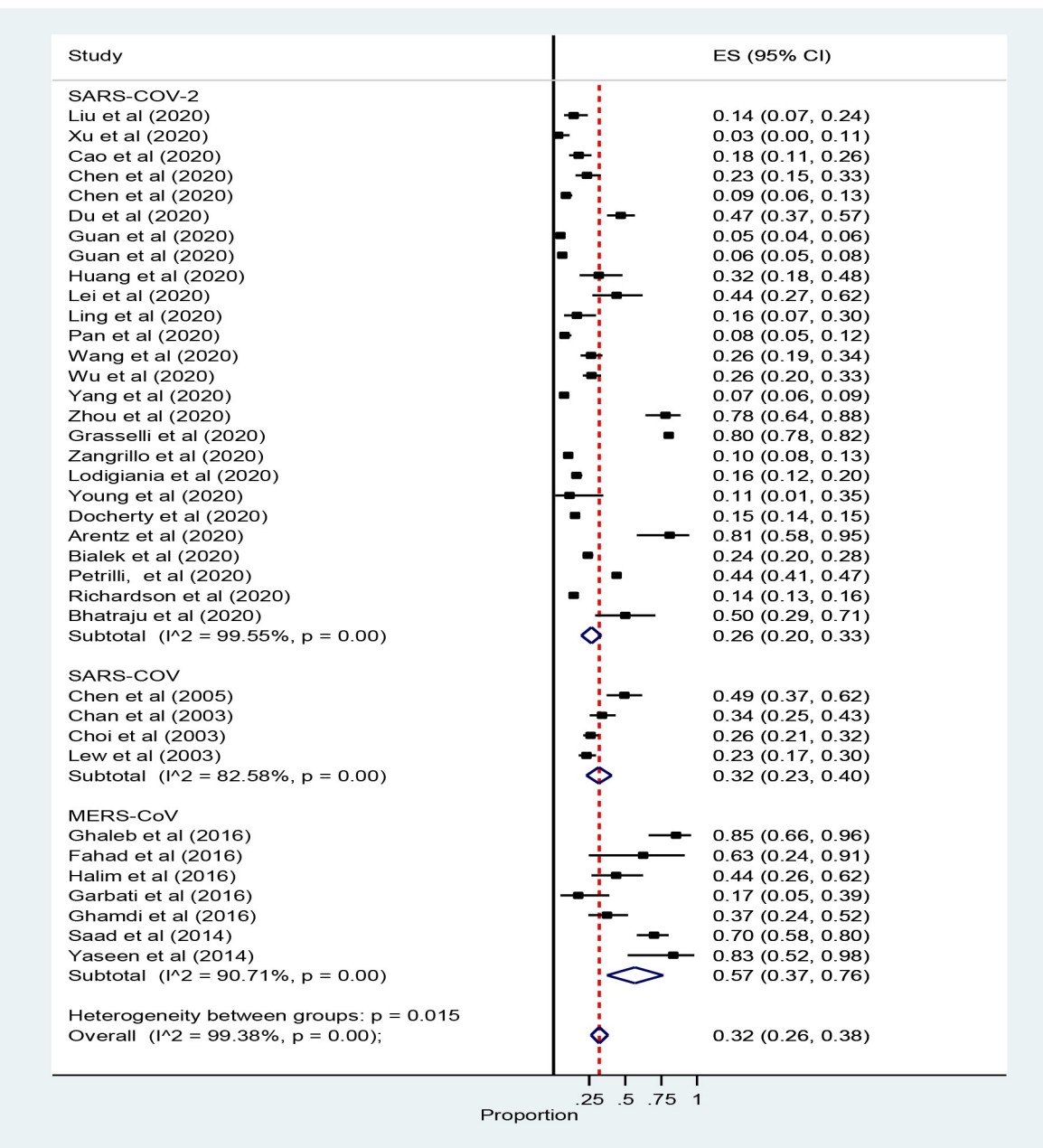

**Fig 3. Forest plot for subgroup analysis prevalence of ICU admission patients with coronavirus: The midpoint of each line illustrates the prevalence; the horizontal line indicates the confidence interval, and the diamond shows the pooled prevalence.** ICU: Intensive Care Unit.

The finding of the subgroup analysis by types of corona revealed that the rate of ICU admission with SARS-COV, MERS and SARS-COV-2 was 32% (95% CI, 23 to 40), 57% 95% CI, 37 to 76) and 26% 95% CI, 20 to 33) respectively (Fig 3).

**3.3.2. Prevalence of ICU mortality.** The Meta-Analysis showed that the prevalence of mortality among ICU admitted patients with Coronavirus was 39% (95% CI: 34 to 43, 37 studies and 24, 983 participants) (Fig 4).

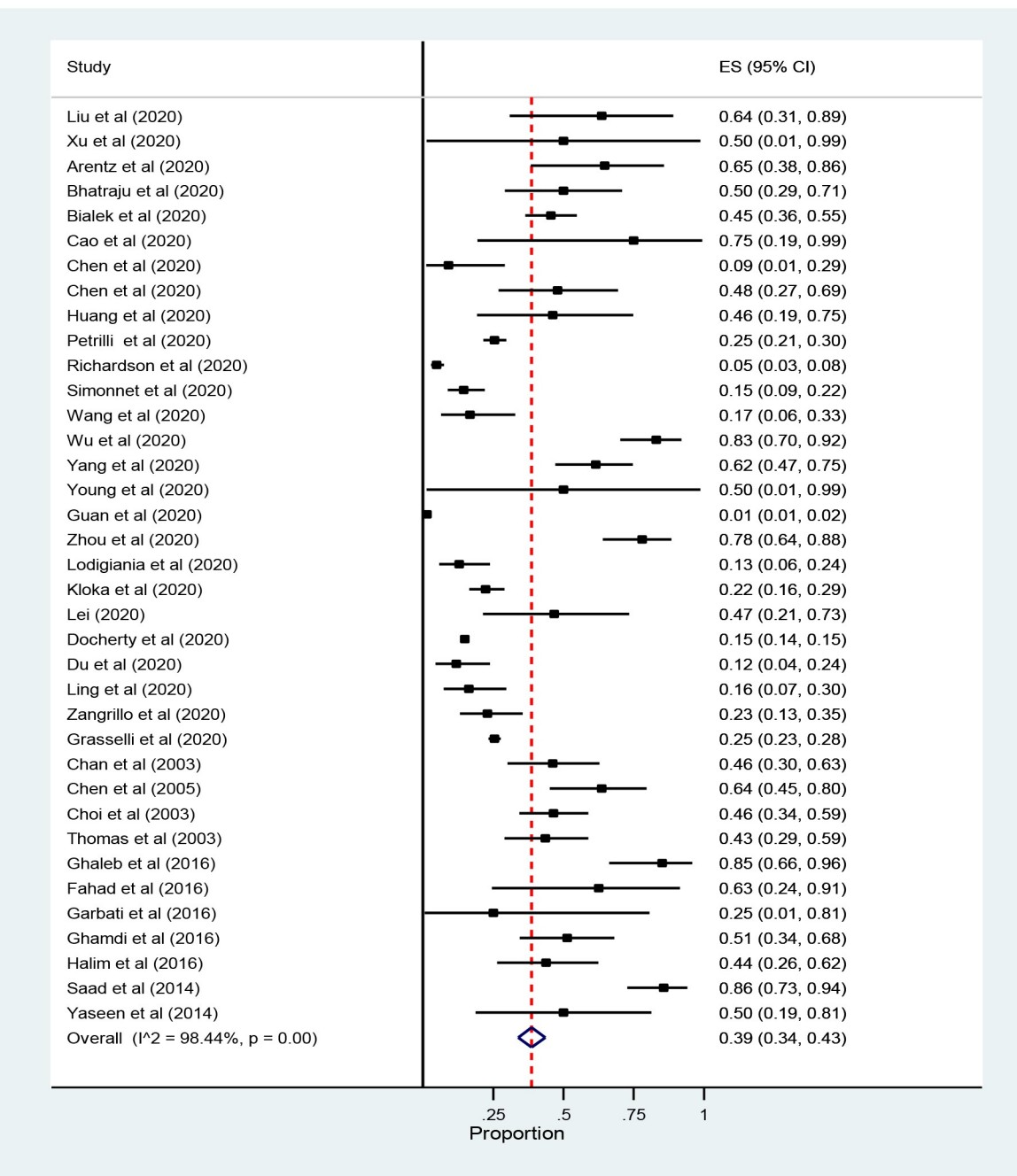

**Fig 4. Forest plot for the prevalence of ICU mortality among patients with coronavirus: The midpoint of each line illustrates the prevalence; the horizontal line indicates the confidence interval, and the diamond shows the pooled prevalence.** ICU: Intensive Care Unit.

The subgroup analysis of the pooled prevalence of mortality among ICU admitted patients with Coronavirus showed that mortality was higher in Saudi Arabia with the Middle East respiratory syndrome 61%(95% CI: 44 to 78) while the prevalence of ICU mortality among patients with the severe acute respiratory syndrome (SARS-CoV-2) was 31% (95% CI: 26to 36) (Fig 5).

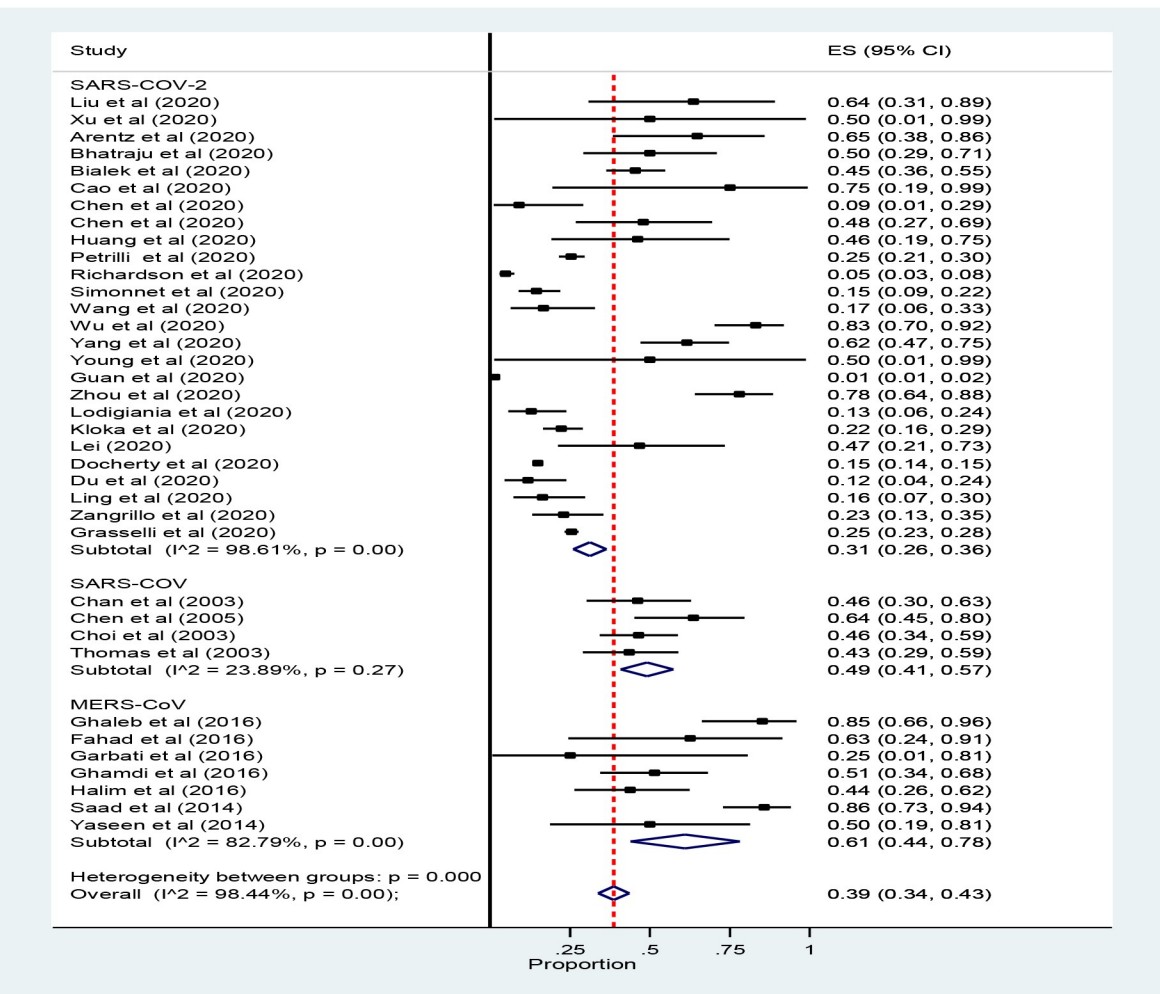

**Fig 5. Forest plot for subgroup analysis of the prevalence of ICU mortality among patients with coronavirus: The midpoint of each line illustrates the prevalence; the horizontal line indicates the confidence interval, and the diamond shows the pooled prevalence.** ICU: Intensive Care Unit.

The subgroup analysis by country revealed that ICU mortality with COVID-19 was 31% (95% CI: 44 to 78, 25 studies, 24677 participants) where the highest was in China 42% (95% CI: 23 to 61, 13 studies, 1480 participants) followed by USA 36% (95% CI: 18 to 53, 5 studies, 992 participants) (S1 Fig).

**3.3.3. Prevalence of comorbidity.** The prevalence of comorbidity among ICU patients with coronavirus was 66% (95% confidence interval (CI): 47 to 85, 12 studies, and 2614 participants) (Fig 6). The Meta-Analysis also revealed that the prevalence of comorbidity among COVID-19 Patients admitted in ICU was 59% (95% confidence interval (CI): 39 to 79, 10 studies and 896 participants) (S2 Fig).

The subgroup analysis by the types of comorbidity showed that cardiovascular diseases were the most prevalent 55% (95% confidence interval (CI): 46 to 64) followed by hypertension and Diabetes Mellitus, 38% (95% confidence interval (CI): 26 to 55) and 31% (95% confidence interval (CI): 20 42) respectively (Fig 7).

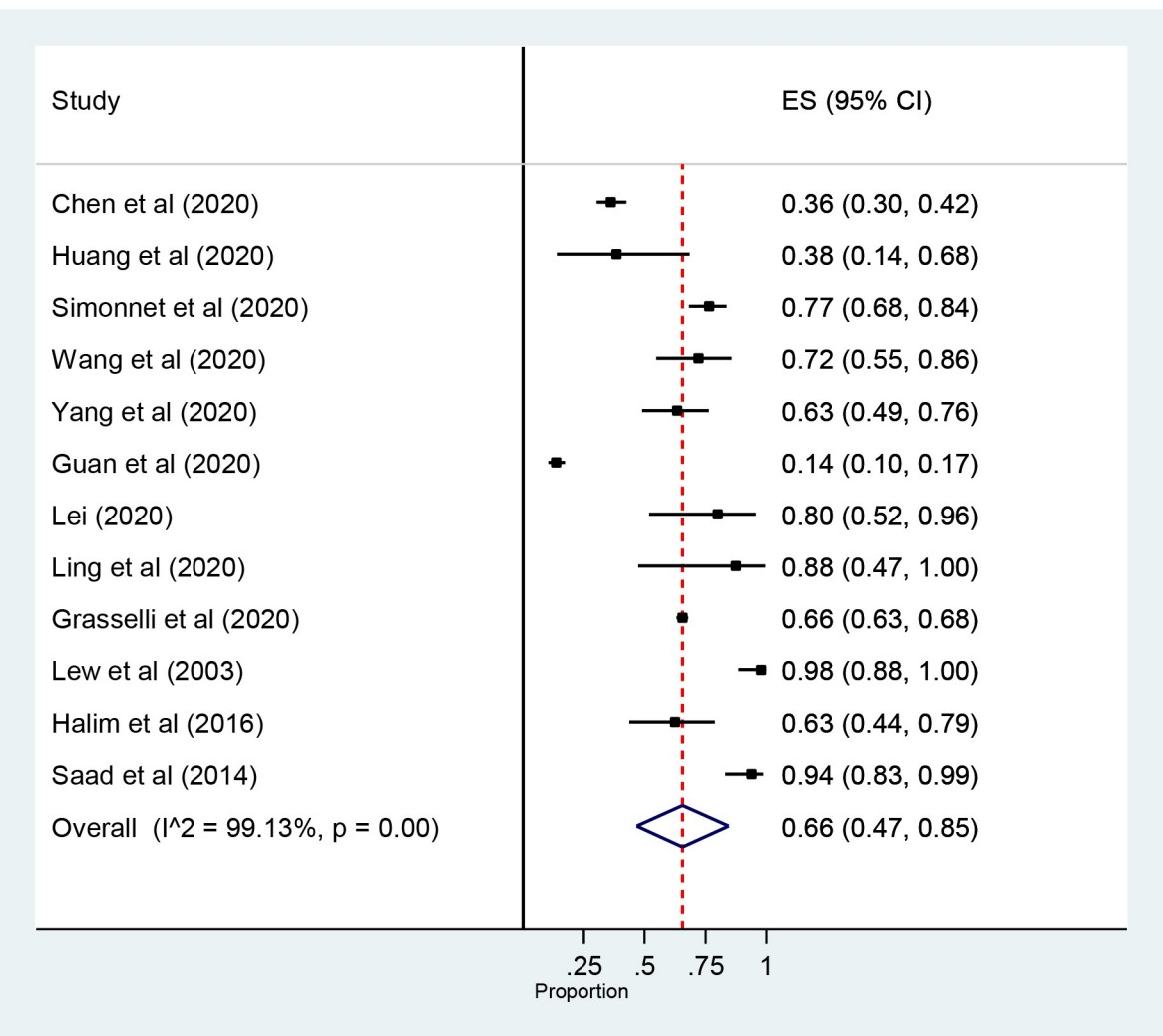

**Fig 6. Forest plot for the prevalence of ICU Comorbidity among patients with coronavirus: The midpoint of each line illustrates the prevalence; the horizontal line indicates the confidence interval, and the diamond shows the pooled prevalence.** ICU: Intensive Care Unit.

### 3.4. Prevalence of complications

The Meta-Analysis showed that the prevalence of complications among ICU admitted patients with coronavirus was 68% (95% confidence interval (CI): 33 to 104) (Fig 8). The subgroup analysis by types of complication showed that ARDS was the most prevalent complication, 54% (95% confidence interval (CI): 26 to 82) followed by infection and sepsis, 47% (95% confidence interval (CI): 29 to 65) and 37% (95% confidence interval (CI): 26 to 49) respectively (S3 Fig).

### 3.5. Regression analysis

The prevalence of mortality among patients with Coronavirus was greatly affected by several factors including the presence of co-morbidities, history of smoking, history of substance use, male gender, older age groups, ICU admission, nosocomial infection, and others. The

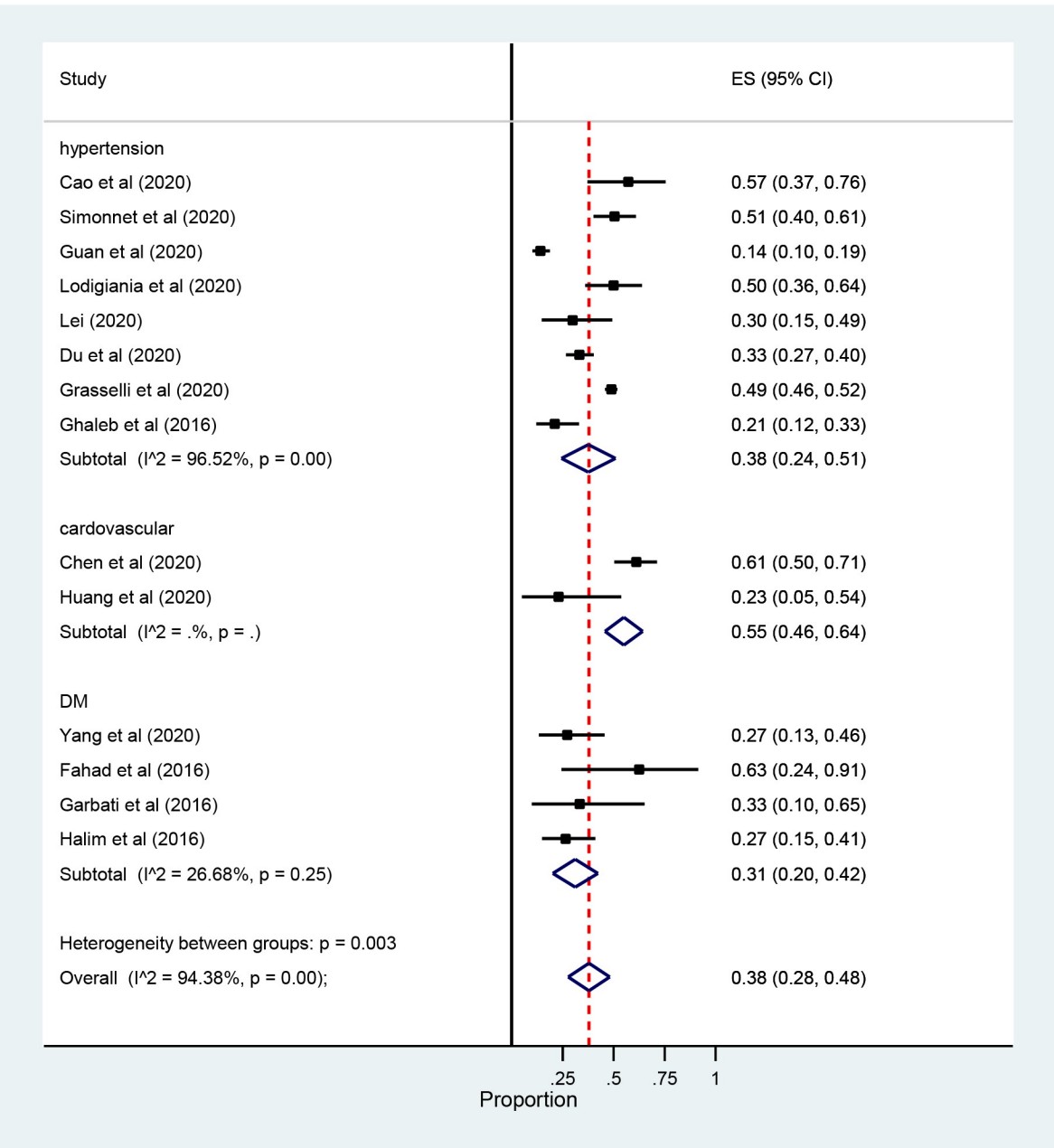

**Fig 7. Forest plot for subgroup analysis of the prevalence of ICU Comorbidity among patients with coronavirus: The midpoint of each line illustrates the prevalence; the horizontal line indicates the confidence interval, and the diamond shows the pooled prevalence.** ICU: Intensive Care Unit.

regression analysis revealed that patients with ARDS were 2 times more likely to die as compared to those who didn't develop ARDS, RR = 2.08 (95% confidence interval(CI): 1.48 to 2.93). The risk of mortality among patients who are older than 50 years increased by 13%, RR = 1.87(95% confidence interval (CI): 1.35 to 2.58). The presence of any comorbidity increased the risk of death by 39%, RR = 1.61(95% confidence interval (CI): 1.24 to 2.09) (S4 Fig).

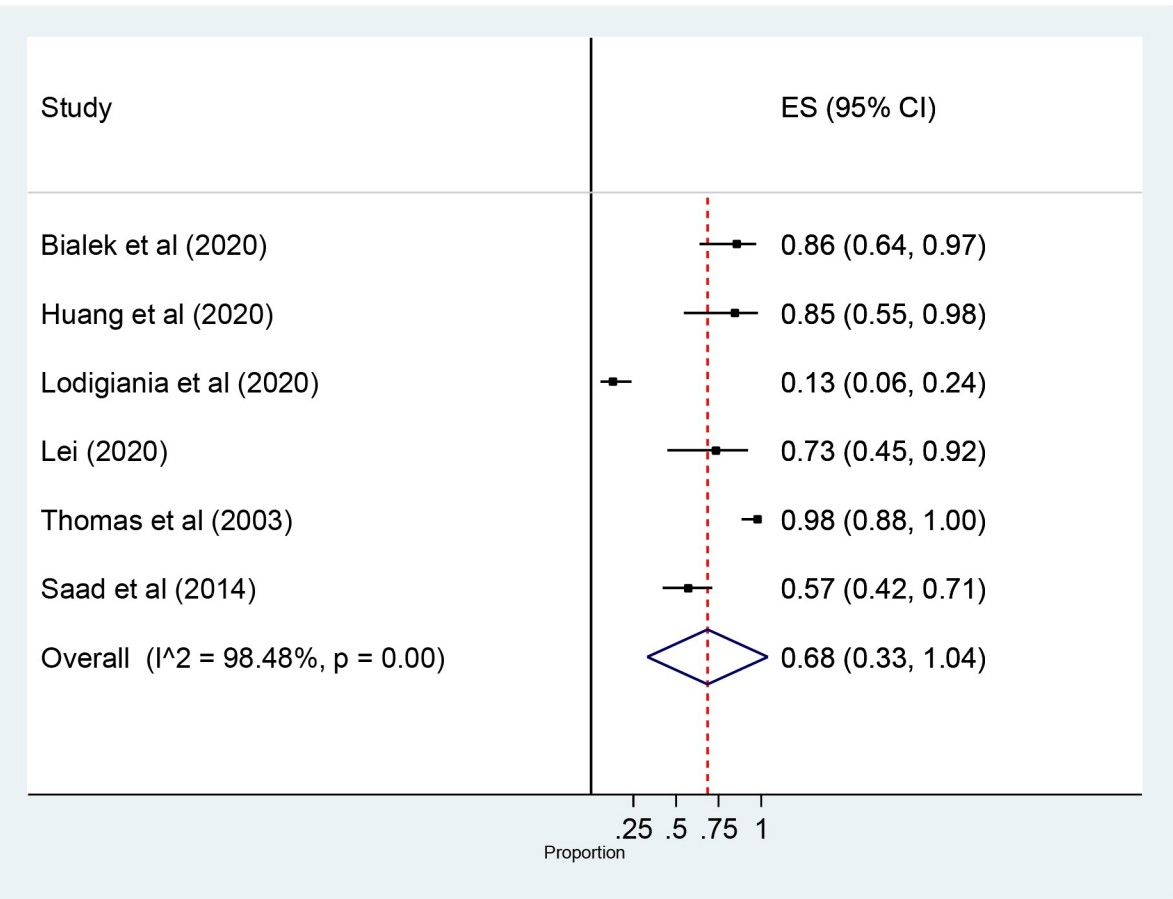

**Fig 8. Forest plot for of prevalence of ICU Complication among patients with coronavirus: The midpoint of each line illustrates the prevalence; the horizontal line indicates the confidence interval, and the diamond shows the pooled prevalence.** ICU: Intensive Care Unit.

## 3.6. Sensitivity analysis and publication bias

Sensitivity analysis was conducted to identify the most influential study on the pooled summary effect and we didn't find significant influencing the summary effect.

Publication bias was investigated with funnel plot asymmetry and egger's regression and Begg's rank correlation were run to investigate publication bias objectively. The funnel plot didn't show significant publication bias. Neither egger's regression nor Begg's rank correlation showed significant publication bias (P-value < 0.1464) (Fig 9).

## 4. Discussion

The Meta-Analysis revealed that more than one-third of patients with coronavirus infection were admitted to ICU globally. The subgroup analysis showed that the rate of ICU admission was very high in patients with the Middle East respiratory syndrome (MERS-CoV), 57% (95% CI: 37to 76) as compared to severe acute respiratory syndrome (SARS-CoV-2 and SARS-CoV), 26% (95% CI: 20 to 33) and 32% (95% CI: 23 to 40) respectively. Currently, the total confirmed cases and the death of patients with the SARS-CoV-2 virus is unpredictably high as compared to the previous two outbreaks [13–15, 19, 20, 63, 64, 66–68]. The lower rate of ICU admission in patients with COVID-19 in this systematic review and Meta-Analysis might be due to a

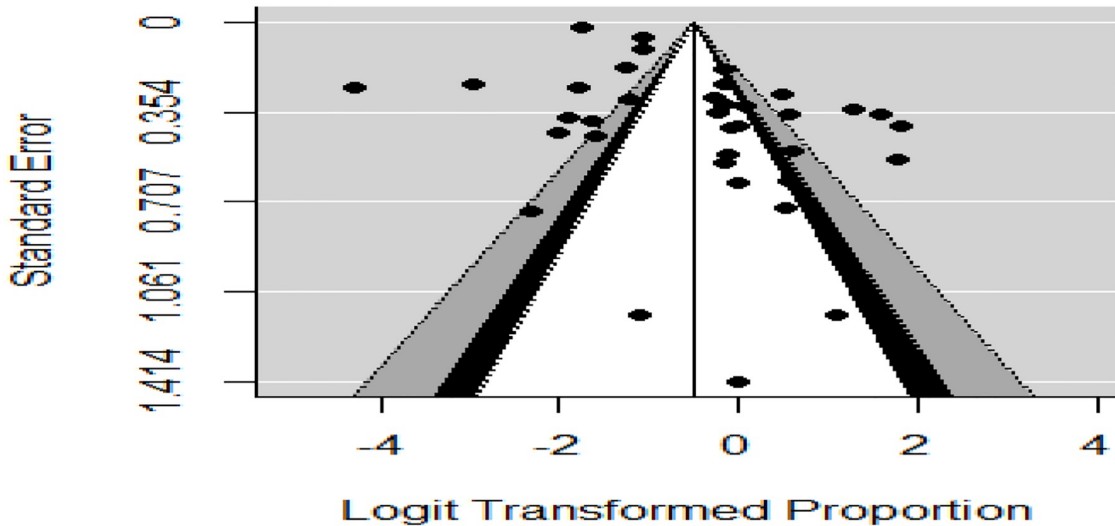

**Fig 9. Funnel plot to assess publication bias.** The vertical line indicates the effect size whereas the diagonal line indicates the precision of individual studies with a 95% confidence interval.

small number of studies assessing rates of admission compared to the number of cases and also the majority of studies were case series with small sample size.

This systematic review and Meta-Analysis revealed that the prevalence of mortality among Coronavirus confirmed cases admitted in ICU were, 39% (95% CI: 34 to 43). This finding is interpreted as there is one mortality for every three cases of admission. This finding is in line with individual studies conducted among Coronavirus confirmed cases since the first outbreak in 2002, China [13–15, 19, 20, 63, 64, 66–68]. The possible explanation for a high number of deaths in ICU may be explained in terms of a limited number of mechanical ventilators, adequate laboratory investigation, integrated patient monitors, presence of co-morbidities, hospital-acquired infections, and some others.

The subgroup analysis showed that the prevalence of mortality among COVID-19 patients admitted in ICU was very higher, 31% (95% CI: 26 to 36). But, it is relatively low as compared to MERS-CoV and SARS-CoV, 61% (95% CI: 44 to 78), and 49% (95% CI: 41 to 57) respectively. The possible explanation for the lower prevalence of mortality among COVID-19 patients might be due to better ICU supportive management, skilled ICU professionals, integrated patient monitors, and lessons from previous outbreaks in handling ICU cases.

The pooled prevalence of comorbidity among patients with coronavirus was as high as sixty percent. The subgroup analysis revealed that the prevalence of comorbidity among COVID-19 patients was 59% (95% confidence interval (CI): 39 to 79) which is consistent with findings of subgroup analysis of SARS-COV, MERS-COV, and individual included studies. The regression analysis revealed that presences of comorbidity, male gender, age greater than 50 years, and ARDS were independent predictors of mortality among patients admitted in ICU with coronaviruses.

## 4.1. Quality of evidence

The systematic review and meta-analysis included plenty of studies with adequate sample size. The methodological quality of included studies was moderate to high quality as depicted with Joanna Briggs Institute assessment tool for meta-analysis of observational studies. However,

substantial heterogeneity associated with dissimilarities of included studies in sample size, design, and location could affect the allover quality of evidence.

## 4.2. Limitation of the study

The review incorporated plenty of studies with a large number of participants but the majority of studies included in this review didn't report data on comorbidity and risk factors to investigate the independent predictors. Besides, there were a limited number of studies in some countries and it would be difficult to provide conclusive evidence with results pooled from fewer studies.

## 4.3. Implication for practice

Body of evidence revealed that rate of ICU admission; the prevalence of mortality; morbidity and complications were very high among patients with COVID-1. These could be a huge impact particularly for low and middle-income countries with a limited number of ICU beds, mechanical ventilator, integrated patient monitor, skilled professionals combined with malnutrition, and communicable disease. Therefore, a mitigating strategy is required by different stakeholders to combat the catastrophic impacts of COVID-19 pandemic through creating awareness about preventive measures, implementing ICU protocols for supportive management, management of comorbidities, and prevention of complications.

## 4.4. The implication for further research

The meta-analysis revealed that the prevalence of mortality among COVD-19 in ICU was very high and the major independent predictors of mortality were identified. However, the included studies were too heterogeneous, and cross-sectional studies also don't show a temporal relationship between mortality and its determinants. Therefore, further observational and randomized controlled trials are in demand for a specific group of patients by stratifying the possible independent predictors.

## 5. Conclusion

The systematic review and Meta-Analysis revealed that approximately one-third of patients admitted to ICU with severe Coronavirus disease. The systematic review also showed that more than thirty percent of patients admitted in ICU with a severe form of COVID-19 for better care died which warns the health care stakeholders to give attention to intensive care patients admitted with COVID-19 through accessing mechanical ventilators, integrated patient monitors, skilled ICU staffs, creation of awareness about infection prevention and more others. Besides, the prevalence of mortality had a strong relation with comorbidity, age, gender, and complication.

## Supporting information

**S1 Table. Description of excluded studies with reasons.**
(DOCX)

**S2 Table. Methodological quality of included studies.**
(DOCX)

**S1 Fig. Forest plot for subgroup analysis of prevalence of ICU mortality by country: The midpoint of each line illustrates the prevalence; the horizontal line indicates the**

**confidence interval, and the diamond shows the pooled prevalence.** ICU: Intensive Care Unit.
(DOCX)

**S2 Fig. Forest plot for subgroup analysis of prevalence of ICU comorbidity by types of coronavirus: The midpoint of each line illustrates the prevalence; the horizontal line indicates the confidence interval, and the diamond shows the pooled prevalence.** ICU: Intensive Care Unit.
(DOCX)

**S3 Fig. Forest plot for subgroup analysis of prevalence of ICU Complication among patients with coronavirus: The midpoint of each line illustrates the prevalence; the horizontal line indicates the confidence interval, and the diamond shows the pooled prevalence.** ICU: Intensive Care Unit.
(DOCX)

**S4 Fig. Forest plot showing pooled odds ratio (log scale) of the associations between Intensive Care Unit mortality and its determinants (A: Co-morbidities; B: Age greater than 50 years; C: Gender D: ARDS).**
(DOCX)

**S1 Checklist. PRISMA checklist.**
(DOC)

## Acknowledgments

The authors would like to acknowledge Dilla University for technical support and encouragement to carry out the project.

## Author Contributions

**Conceptualization:** Semagn Mekonnen Abate, Bivash Basu.

**Formal analysis:** Semagn Mekonnen Abate, Siraj Ahmed Ali, Bahiru Mantfardo, Bivash Basu.

**Methodology:** Semagn Mekonnen Abate.

**Project administration:** Semagn Mekonnen Abate.

**Writing – original draft:** Semagn Mekonnen Abate, Siraj Ahmed Ali, Bahiru Mantfardo, Bivash Basu.

**Writing – review & editing:** Semagn Mekonnen Abate, Siraj Ahmed Ali, Bahiru Mantfardo, Bivash Basu.

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
