## [Decision Letter · Decision Letter 0]

8 Jun 2020

PONE-D-20-10711

Rate of Intensive Care Unit admission and outcomes among patients with coronavirus: A systematic review and Meta-analysis

PLOS ONE

Dear Dr. Mekonnen,

Thank you for submitting your manuscript to PLOS ONE. After careful consideration, we feel that it has merit but does not fully meet PLOS ONE’s publication criteria as it currently stands. Therefore, we invite you to submit a revised version of the manuscript that addresses the points raised during the review process.

We look forward to receiving your revised manuscript.

Kind regards,

Chiara Lazzeri

Academic Editor

PLOS ONE

Journal Requirements:

2. Please confirm that you have included all items recommended in the PRISMA checklist including details of reasons for study exclusions in the PRISMA flowchart and number of studies excluded for each reason.

3. Please confirm that you have included all items recommended in the PRISMA checklist including the full electronic search strategy used to identify studies with all search terms and limits for at least one database.

4. We suggest you thoroughly copyedit your manuscript for language usage, spelling, and grammar. If you do not know anyone who can help you do this, you may wish to consider employing a professional scientific editing service.  

'No funding was obtained from any organization'

'No'

7. Thank you for stating the following in your Competing Interests section: 

'No'

8. Please amend either the abstract on the online submission form (via Edit Submission) or the abstract in the manuscript so that they are identical

Additional Editor Comments (if provided):

. Please do not edit.]

Reviewers' comments:

Reviewer's Responses to Questions

**Comments to the Author**

1. Is the manuscript technically sound, and do the data support the conclusions?

Reviewer #1: Yes

Reviewer #2: Yes

2. Has the statistical analysis been performed appropriately and rigorously? 

Reviewer #1: Yes

Reviewer #2: Yes

3. Have the authors made all data underlying the findings in their manuscript fully available?

Reviewer #1: Yes

Reviewer #2: Yes

4. Is the manuscript presented in an intelligible fashion and written in standard English?

Reviewer #1: Yes

Reviewer #2: Yes

5. Review Comments to the Author

Reviewer #1: In this systematic review, dr. Mekonnen and colleagues present results of a systematic review and meta-analysis of cohort studies investigating prevalence of ICU admission and ICU mortality among patients with coronavirus infection (SARS, MERS, and COVID-19). They found that prevalence of ICU admission is about 16% and mortality among ICU patients is about 50%

Given the current COVID-19 pandemic and the few and sparse data available, the Authors’ work deals with an interesting an up-to-date topic. Nevertheless, I have a few comments that I hope will help the Authors to improve their work.

1. Abstract. Please specify study objective in the background. Please specify primary outcome in the Methods, as well as date of the search.

2. Abstract. Please explain what does “Google scholars up to ten pages” means. I suggest to explain this in the main text, and delete this from the abstract.

3. Abstract. Please add inclusion/exclusion criteria to the abstract

4. Abstract. Please specify the total number of studies identified from the Search strategy, and the total number of studies included

5. Introduction. Please shorten the introduction. Detailed description of SARS, MERS and COVID-19 mortality is not necessary and can be moved to the discussion. Similarly, incidence of COVID-19 in different areas can be moved to the discussion. Finally, also detailed description of various predictors identified can be moved to the discussion

6. Methods. I believe that the Section “Eligibility criteria” contains redundant information. It could all be reported as a clear list of inclusion criteria/exclusion criteria.

7. As a related point, please note that among exclusion criteria there is “studies that didn’t” followed by “traumatic brain injury”. Please correct

8. Methods, study outcomes. Please leave a separate paragraph specifying primary and secondary outcomes

9. As a related point, this reviewer was unable to find data on the secondary outcomes (length of ICU stay, duration of mechanical ventilation, secondary infections) in the meta-analysis. Please report these data or clearly state that no studies reported this information

10. The description of the search strategy is unclear. In particular, it is unclear to me what does “Google scholar up to ten pages” means. Please report the keywords used for search strategy in the supplementary appendix

11. Please specify in the methods which subgroup analyses were performed and which were pre-planned

12. Please specify in the methods how was study quality assessed. Please clearly describe items evaluated when assessing study quality

13. I suggest to perform a sensitivity analysis including only high-quality studies

14. Results. I suggest to divide the Results section in clear subsections: 1) study characteristics including study quality 2) primary outcome (including meta-analysis), 3) secondary outcomes 4) subgroup analyses 5) effect of comorbidities on outcome

15. As a related point, please leave comments on the results for the discussion (e.g. “mortality admitted to the ICU was very high”)

16. Please note that Begg’s and Egger’s test should have p-values, while funnel plot is a figure. Please report p-values for Begg’s and Egger’s test

17. Please expand the discussion, and divide it into the following sections: 1) key findings 2) relationship with previous studies 3) implications of study findings for current practice/literature 4) future studies/future directions 5) strength and limitations 6) conclusions

18. Please double check the reference list, to ensure that references are in the journal’s style.

Reviewer #2: This paper by Mekonnen et al attempts to do systematic review on ICU mortality rates among patients verified with infection of coronavirus. The review is organised in accordance to PRISMA criteria and follows as such guidelines for systematic review. Using on line search for relevant journal several papers have been identified. In accordance to PRISMA flow chart twenty two studies were included for review. The authors document average ICU mortality at 50% for patients with coronavirus infection.

The study is nicely organised and authors deserve credit for the effort done to bring attention on the highly morbid disease when treated in ICU. In intro authors do great job to describe current status of covid pandemic although data already seems outdated. The aim seems relevant however this referee would prefer its focus being narrowed. Would it be possible to highlight the troubling low incidence of covid in africa? Discussion starts ok but it is rather thin just to nail high mortality rates without providing scientific arguments for why it is so. The knowledge on covid has expanded massively and what authors wrote yesterday is probably outdated tomorrow. However please provide info on why mortality in different parts of the world may be different. In addition it would be highly relevant for more focus on situation in african countries. I think such would bring the paper to PLOS One upper level.

6. PLOS authors have the option to publish the peer review history of their article (what does this mean?). If published, this will include your full peer review and any attached files.

Reviewer #1: No

Reviewer #2: No

---

## [Author Response · Author response to Decision Letter 0]

16 Jun 2020

These part was uploaded in manuscript tracking labelled " response to reviewers."

As it was pointed out by one of reviewer, we felt that the information provided about COVID-19 were outdated. We updated our search and additional 15 studies with a total of 37 studies were included. All the comments provided were very important and we took them as it is and tried to address section by section as we tried to display in response to reviewers document.

however, we kept description of some epidemiology and mortality data in background section as we feel the background looked shallow and incomplete. 

We thank you very much for your valuable comments 

wishing you all the best!!!

---

## [Editor Report · Decision Letter 1]

22 Jun 2020

Rate of Intensive Care Unit admission and outcomes among patients with coronavirus: A systematic review and Meta-analysis

PONE-D-20-10711R1

Dear Dr. Mekonnen,

We’re pleased to inform you that your manuscript has been judged scientifically suitable for publication and will be formally accepted for publication once it meets all outstanding technical requirements.

Kind regards,

Chiara Lazzeri

Academic Editor

PLOS ONE
---

## [Editor Report · Acceptance letter]

24 Jun 2020

PONE-D-20-10711R1 

Rate of Intensive Care Unit admission and outcomes among patients with coronavirus: A systematic review and Meta-analysis 

Dear Dr. Mekonnen:

I'm pleased to inform you that your manuscript has been deemed suitable for publication in PLOS ONE. Congratulations! Your manuscript is now with our production department. 

Kind regards, 

on behalf of

Dr. Chiara Lazzeri 

Academic Editor

PLOS ONE